# Towards Artificial Intelligence Applications in Next Generation Cytopathology

**DOI:** 10.3390/biomedicines11082225

**Published:** 2023-08-08

**Authors:** Enrico Giarnieri, Simone Scardapane

**Affiliations:** 1Cytopathology Unit, Department of Clinical and Molecular Medicine, Sant’Andrea Hospital, Sapienza University of Rome, Piazzale Aldo Moro 5, 00189 Rome, Italy; 2Department of Information Engineering, Electronics and Telecommunications, Sapienza University of Rome, Via Eudossiana 18, 00196 Rome, Italy; simone.scardapane@uniroma1.it

**Keywords:** cytopathology, digital pathology, artificial intelligence, machine learning, metaverse, natural language processing, blockchains

## Abstract

Over the last 20 years we have seen an increase in techniques in the field of computational pathology and machine learning, improving our ability to analyze and interpret imaging. Neural networks, in particular, have been used for more than thirty years, starting with the computer assisted smear test using early generation models. Today, advanced machine learning, working on large image data sets, has been shown to perform classification, detection, and segmentation with remarkable accuracy and generalization in several domains. Deep learning algorithms, as a branch of machine learning, are thus attracting attention in digital pathology and cytopathology, providing feasible solutions for accurate and efficient cytological diagnoses, ranging from efficient cell counts to automatic classification of anomalous cells and queries over large clinical databases. The integration of machine learning with related next-generation technologies powered by AI, such as augmented/virtual reality, metaverse, and computational linguistic models are a focus of interest in health care digitalization, to support education, diagnosis, and therapy. In this work we will consider how all these innovations can help cytopathology to go beyond the microscope and to undergo a hyper-digitalized transformation. We also discuss specific challenges to their applications in the field, notably, the requirement for large-scale cytopathology datasets, the necessity of new protocols for sharing information, and the need for further technological training for pathologists.

## 1. Introduction

Cytopathology is a branch of laboratory medicine that studies details of cellular morphology useful for cancer screening and early diagnosis. Compared to histopathology, cytology focuses on specific pathological features of single cells in a context of thousands of cells in a specific tissue architecture. Modification of cell properties and morphology reflect the biological status of a specific organ [1,2,3]. Cellular material is taken by using exfoliative cytology, body fluids, scraping, and aspiration cytology, and its morphological aspects are used to formulate a diagnosis using internationally recognized guidelines [4,5,6,7,8,9]. In diagnostic cytopathology it is expected that cytologists scrutinize every cell under the microscope or in gigapixel whole slide images to search for alterations, which are sometimes represented only in a few groups of cells. This can represent a challenge for the cytologists, involving highly time-consuming work and tediousness [10]. Technology involving artificial intelligence (AI) has shown remarkable progress in medicine, including image interpretation and computer assisted diagnosis both in histopathology and cytopathology. Although this process is considered in an early stage, it probably will represent the third revolution in pathology, through the introduction of AI in medical routines in which pathologists will be central to the development of algorithms and their validation [11,12]. Deep learning, as a branch of machine learning and as the major tool in the current AI wave, has greatly accelerated the development of computational cytopathology-exploiting algorithms and specific architecture designs such as multilayer perceptrons (MLP), convolutional neural networks (CNN), recurrent neural networks (RNN), and transformers [13]. In addition, in the last decades spatial computing and the metaverse have also been empowered by AI. Their synergy can create a new scenario for digital pathology in both teaching and diagnosis through platforms, devices, chatbots, and other human–machine interaction tools. In this perspective paper, we will review these advances in machine learning techniques and evaluate practical aspects for their application to digital cytopathology, including future developments and open challenges.

The paper, while based upon the experience and knowledge of the authors, provides an entry point both for pathologists interested in how AI technologies will impact the field, and for AI practitioners who want to gain a perspective on specific challenges and opportunities of this new wave of applications in the medical domain. The paper is organized as follows. In the rest of this section, we provide a brief historical perspective on the use of automation techniques in cytopathology, starting with early diagnostic systems in the 1960s up to today. In Section 2 we describe deep learning models for computer vision and classical applications in the medical field, including object detection (e.g., cell counting) and segmentation, along with some specific challenges, such as the need of improving data acquisition and quality. We then overview the use of AI-powered technologies, including virtual reality for training and visualization (Section 3), natural language processing (Section 4), and decentralized technologies (Section 5). We conclude in Section 6 with some additional comments and a summary.

### From Cytology Automation to Artificial Intelligence

In cytopathology, screening for the early detection of cervical cancer was one of the largest early applications of image analysis, through the construction of platforms using microscope units, software tools for display, and tele-control. These early systems showed limitations but also advantages. In 1952, at the University of Tenneesee, Mellors et al. designed the Cytoanalyzer, the first semi-automated screener based on an optical electronic machine to speed up detection of cancer cells of the uterine cervix. The application of the Cytoanalyzer was intended to reduce the scarcity of technicians to analyze cells and improving early diagnosis of uterine cancer [14]. In the mid 1960s, Taxonomic Intra-Cellular Analytic System (TICAS) demonstrated utility in the field of automated diagnostic systems [15]. In the 1970s, Zahniser et al. developed the BioPER system, a sophisticated software to obtain a high throughput of smears per hour with a low percentage of false positives and false negatives, and they introduced a fixed cutoff of 2% “abnormal” cells on a slide to trigger an alarm [16]. By 1989, hardware and software were improved allowing systems like Leytas, Cytopress, Cervifip, and Cyto-Savant to reduce the workload, screening time, and errors through more interactive diagnostic procedures [17,18,19]. A new approach to cell classification began in the 1980s with neural network technology and the popularization of the backpropagation training algorithms applied in many areas, including cytology automation. The first commercial approach using artificial intelligence (AI) in cytopathology was the PAPNET^TM^, a semiautomated system based on neural network modeling. This system was introduced for quality control in smear rescreening, leaving the decision directly to the machine, through internal algorithms. The system was aimed at reducing the number of false negatives and was an additional tool for the interpretation of abnormal cells [20,21]. The interest in neural networks was renewed after 2012, with deep neural networks (DNNs) becoming the state-of-the-art solution for multiple benchmarks in the computer vision and natural language processing fields [22,23] and, more recently, with the emergence of large language models (LLMs) such as ChatGPT. DNNs demonstrated an ability to work directly on raw images [24], and they can be trained to classify, segment, and process images with extremely high accuracy in a variety of fields. Consequently, investigations have started into the use of DNNs in several medical applications, including diagnostic cytology [25].

## 2. Applications of Computer Vision Models to Cytopathology

As stated before, advances in machine learning have recently impacted cytopathology, providing opportunities for all pathologists in their daily work. Two important computer tasks in this context are detection and segmentation. Detection is the task of finding specific objects in an image, such as neoplastic cells in the context of normal cells. Machine learning can be used to classify the grade of atypia for each single cell by highlighting them with the proper bounding box, which requires specialized object detection networks (Figure 1). Segmentation involves categorization of each pixel in the image with a specific class, allowing a fine-grained separation of the cells from their background. While patch-based convolutional neural networks can identify and locate objects of different types, segmentation detects not only objects but also their boundaries without suffering from different staining conditions or hand-crafted features, resulting in an important tool in whole slide imaging [26,27,28,29,30,31,32]. In their daily routine, cytopathologists analyze and integrate a large amount of morphological information. Hundreds of thousands of different cell features are simultaneously examined by a human mind skilled at quick interpretation. Furthermore, modern cytology increasingly integrates clinical information, immunocytochemical staining, and molecular pathological data, especially in diagnostically difficult cases and when clinicians require prognostic factors.

From this point of view, automatic ways of cell counting, boundary identification, and cell classification in digital pathology are seen with great expectations, although there are still challenges. Cytopathologists must learn how to use algorithms correctly (including their drawbacks, described below), how they work and, above all, their clinical utility [33]. The combination of image analysis and machine learning (ML) could be the key to improving the quality assurance, reducing factors that can cause diagnostic errors. This approach would require laboratories to be equipped with specific technologies and skilled staff.

### 2.1. Data Acquisition and Availability

The creation of digital slide libraries, now available on different public or private platforms, is rapidly transforming digital pathology [34]. In the future, each laboratory will develop its own dataset of images, classified by type of disease, to be shared with other laboratories for educational and diagnostic purposes and to develop algorithms. To this end, operators must learn how to use novel annotation software for images (e.g., the VGG Image Annotator developed by the Visual Geometry Group) [35], and to use specialized deep neural networks (DNN) models such as convolutional neural networks (CNNs). Several convolutional neural network architectures are available to process images including medical images. EfficientNets, MobileNet, XceptionNet, and InceptionNetv3 architectures demonstrated accuracy, model’s efficiency, and low computational costs [36,37,38,39]. In cytopathology, there is an elevated level of complexity due to sample preparation types, the presence of hypercellularity with the multitude of cytologic substrates, and similarity of morphological features. These aspects requires complex mental reasoning based on a pathologist’s experience of a large data set of images. This means that the training phase must be carried out using thousands of high-quality images, each annotated by an expert pathologist, which would involve a considerable amount of time and work, to ensure that a trained network can effectively generalize across different scenarios, equipment, and laboratories. In the future, this could be solved through decentralizing image banks in various institutions and making them available through a blockchain-based network or federated learning (Section 5), or by using self-supervised algorithms.

### 2.2. Current Challenges and Limitations

Despite their promising performance, DNNs applications have limitations that must be acknowledged by pathologists. Firstly, they still require a large amount of expertly labelled data to be trained, especially in medicine and pathology. Fields that have such data publicly available benefit more than do fields for which this training is still ongoing, including cytology. In computer vision, this problem has been tackled by the emerging field of self-supervised learning, which allows the pre-training of neural network models (sometimes known as “foundation models”) using large sets of unlabeled images before tackling a downstream task, such as segmentation, where few labeled points are known. While some initial progress has been made in the development of foundation models for medical imaging, this is still an open challenge for cytopathology [40,41]. Secondly, DNNs are “black box” classifiers, meaning that it is generally difficult to understand why a certain image has been classified in a certain way [42]. For this reason, more recent works have sought to integrate the predictions of DNNs with techniques capable of improving interpretability and understanding by physicians who are not experts in algorithms and artificial intelligence [43,44]. However, we underline that most applications of explainability techniques today require users to be proficient in the AI models themselves. Developing explainability tools for clinicians or doctors with limited knowledge of neural networks, evaluating them in a real-world setting, and integrating them in production environments are still open challenges [45]. Third, when looking at the confidence scores in output, and not just the most probable class, most neural networks tend to be overly confident and uncalibrated, i.e., the predicted probabilities tend to underestimate the true probability of error. This is a major problem when the confidence in the output must be used in a clinical process to carefully evaluate cost-benefit trade-offs [39]. An uncalibrated model can indeed provide unrealistic confidence in certain predictions, which in turn can be problematic in a medical setting where important diagnostic decisions must be made [46].

### 2.3. Improving Data Acquisition and Quality

To date, digital pathology images have been obtained with devices such as microscopic cameras or slide scanners. These devices cannot make completely identical digital images, even when the image is taken using the same microscope and camera sequentially. Ogura et al. reported discordant classification results between paired digital histopathology images obtained from two independent scans using the same microscope [47]. Compared to thin prep slide preparations, conventional cytology is generally much thicker, resulting in patches of cells defocused when examined under the microscope, and this requires pathologists to change focal plane continually. Recently, deep learning methods have been reported to increase accurate cellular quantification, higher image sharpness, and the number of image details using the dual-view system compared to single-view imaging. Furthermore, defocusing problems can be addressed using domain normalization net (DNN) and refocusing net (RFN) methods to improve data set performance from cervical cytopathology images [48,49]. Overall, we expect neural networks will continue to have a significant impact in improving the data acquisition process in cytopathology laboratories, similar to the role they have on faster MRI acquisition or X-ray diagnostics.

## 3. Use Cases for Augmented and Virtual Reality in Cytopathology

AI-powered emerging technologies such as augmented reality (AR)/virtual reality (VR) and the metaverse can potentially create a realistic virtual world to support learning and diagnosis in digital pathology and cytopathology. With high-bandwidth 5G, and in the future 6G, ML and neural network models will become ever more widespread for different tasks and in different contexts. Through human–machine interaction tools with immersive technologies like head-mounted displays supported by AI, it will be possible for the pathologist to view whole slides in a metaverse environment and easily interact with one or more remote colleagues. However, in the virtual world there are still some technical challenges to solve, such as image quality reduction, noise, haze, blurring, and low resolution that can influence visual perception. Some preliminary CNN architectures were proposed to reduce these issues [50,51,52]. Compared to AR and VR, mixed reality (MR) has demonstrated potential utility in the metaverse due to its hybrid physical–virtual experiences, delivered via two main types of devices: holographic and immersive. In the first case, holographic technology offers the possibility of manipulating physical objects, allowing users to interact with virtual objects in a virtual world. Mixed reality technologies demonstrate many healthcare benefits when integrated with tools for preparing surgical sites [53] or for viewing whole slide images in a virtual environment [54]. To move cytopathology into a virtual scenario, specifically from an educational point of view, the technical challenges and human adaptability should be taken into consideration (Figure 2).

Currently, VR technology available for digital slide navigation does not acquire images in 3D and imaging tools do not fulfill all the requirements for fast and high-resolution acquisition. VR can be improved by the creation of a virtual projection of 2D images in a simulated 360° environment. For example, GANverse3D, introduced by NVIDIA, transforms 2D images into 3D animated objects that can be viewed and controlled in virtual environments within Nvidia Omniverse [55]. However, seeing mixed reality content such as 3D holograms will need 5G technology that can transfer data in a huge bandwidth within the shortest time possible, integrating mixed reality in medical devices or image records into holograms compatible with devices. Finally, participants without sufficient experience and time spent in the VR environment show well-documented side effects such as nausea, eyestrain, and seizures; therefore, long-term usage of VR in clinical practice deserves further investigation [56]. Summarizing these points, the next generation of devices must be improved to provide visual–interactive experiences with reduced side effects, costs, and workflow interruption, while maintaining standardization in the imaging process, which are the most important aspects to address to reduce professional reluctance to adopting new technologies. We also note that medical 3D consultation or teaching are already in the experimentation and use phases, a field where VR and AR technologies can have a significant impact. Future tasks for educational use of VR environments in medical training will be characterized by important challenges for medical educators and students. Instructors that want to apply VR environments in medical education need to properly understand each type of technology available, to facilitate student adaptation, avoid negative effects during learning activities, and ensure long-term practicality of human–computer interaction in medical routines in the future.

## 4. Natural Language Processing in Cytopathology

Natural language processing (NPL) concerns the application of statistical, computational, and AI models to process and analyze large amounts of text [57]. Large language models (LLMs) such as the Generative Pretrained Transformer (GPT) have emerged as the main tool in the use of neural networks for NLP. LLMs are trained using a huge amount of textual data, mostly gathered from the internet, using a combination of techniques such as next-token prediction and instruction tuning. They show a surprising level of reasoning and problem-solving capabilities, and they can be used for many different tasks such as language translation, text summarization, and dialogue systems. Importantly, they can answer questions and interact in a conversational fashion, making them accessible also to non-expert users. Although ChatGPT and similar open-source models such as LIMA are currently subject to debates on plagiarism and cheating, in some sectors such as healthcare they could make an important contribution [58,59,60]. For example, for teaching assistance LLMs might be useful to generate exercises, quizzes, and scenarios in the classroom or at home to help practice and aid through a virtual tutor that can answer students’ questions and provide feedback on their progress. In healthcare, there is a list of potentially ideal LLMs tools: virtual assistants for telemedicine in cases of remote patient monitoring, medical education for students and healthcare professionals, research, and clinical trials [61]. In cytopathology, LLMs could be used in teaching and routine diagnostics. In the first case, an LLM could help in the initial theoretical stage of study, helping students to discover bibliographic material, guidelines and to explore basic concepts of cytopathology in different organs. In diagnostics, LLMs could support a discussion about morphological aspects in a specific clinical case, within a forum between professionals and a virtual cytopathologist, in order to choose specific molecular markers to complete a diagnosis. It could facilitate the navigation of vast amounts of medical and pathological information on the internet, compare specific images in large data sets, discover literature reviews summarizing relevant articles, and take part critically in the debate about a possible diagnosis (Figure 3). Recently, large research efforts have been oriented to deep learning-based image captioning through arcitectures capable of processing images and generating language [62]. In cytopathology, it may be of interest to develop a model of image information including text captions. It may help a model to learn morphological features of an image from experts and validated text descriptions, and, therefore, have an automated capacity to output textual interpretations that are putatively consistent with its training data.

### Limits of LLM Models

Unfortunately, the simplicity of interacting with a GPT-like model through a text interface hides the complexity of using it in an efficient way to obtain useful and actionable results. In particular, “prompt engineering” is emerging as a new research direction to find the best ways to elicit good responses from LLMs. For example, including “you are an expert in linear algebra with multiple years of experience” when querying a generalist model like ChatGPT on linear algebra topics can improve the quality of the answers. This means that users will need to be proficient at several emerging techniques which are still evolving in the literature, such as few-shot prompting or user-based fine-tuning, to align the model to their preferences. This creates challenges to their use and may result in models that are sub-optimal for a given task when using out-of-the-box commercial models such as ChatGPT. In addition, using LLM technologies will require users to focus more on data curation and security, to avoid models that can be hijacked to elicit sensible information memorized from their training set, or that replicate (or generate) fake or unclear information [63,64]. Recently, Peng et al. developed a generative LLM, namely GatorTronGPT, for the medical domain to evaluate its utility for research and healthcare. The study used a GPT-3 architecture with 277 billion words of clinical text mixed with English text, demonstrating the utility of synthetic clinical text generation for clinical research with linguistic readability comparable to real world clinical notes [65].

## 5. Decentralized Technologies in Cytopathology

Blockchain is an example of a decentralized data storage facility, that acts as a digital ledger for storage of a list of assets using cryptography technology without a centralized entity [66]. In the last decade, advanced methods that combine decentralized data storage techniques and AI models have been proposed. Cooperation between deep learning and blockchain has proven useful by removing the need to centralize data storage or to control data flow and modifications, and, for this reason, they have several applications, especially in healthcare [67,68]. For example, blockchain allows patients to assign rules for access to their medical data, permitting access to parts of their data for diagnostic consultation or research for a fixed time period. In digital pathology, high-resolution images enable physicians to collect, tag, expand, share, and analyze specific sections of the image slides, reducing the time of diagnosis [69,70]. In recent years, we have observed how AI has become crucial especially in precision medicine, where a DL model has shown potential in the prognosis and diagnosis of cancer using a vast amount of information including gene mutation status, molecular subtypes, microsatellite instability (MSI) related to histopathology in different cancer types, thus, showing important clinical applicability [71,72]. These models are characterized by large and diversified data that are trained by a single server; however, in cases of datasets located in different institutions and countries, where regulations on patient information differ, data sharing may become complicated. Decentralized model solutions to circumvent this issue are federated learning (FL) and swarm learning (SL). In FL, data resides at the original location and only model parameters are shared among participants and, possibly, a centralized orchestrator, during training. Having a federated learning model is the key to exploiting unlabeled data, that will allow multiple, geographically separated institutions to share their data with controls to protect patient privacy, while permitting access to self-supervised algorithms. SL represents a decentralized learning system that combines edge computing, blockchain-based peer-to-peer networking and coordination, preserving confidentiality, privacy, and security without a central coordinator [73,74]. For example, these methods could be applied in a cytopathology laboratory to get a faster diagnosis by using ML, when there is not enough labelled images to train the model. To solve the problem, the cytopathologist can obtain sets of images of a similar case from another laboratory. This may not be possible due to data confidentiality; however, regulations on cytopathological imaging have not been introduced yet. Both FL and SL could resolve the privacy issue.

## 6. Conclusions

In view of its importance in making a correct diagnosis and, thus, in selecting an appropriate course of treatment for the patient, adaptation in the clinical practice of cytopathology has become increasingly important in order to integrate this practice with the latest technological developments in AI, immersive technologies, and decentralized algorithms. When making a diagnosis using a microscope, cytopathologists need to be aware of the possibile integration of helpful AI diagnostic models, 3D modeling tools to interact with scans in a more immersive fashion, and dialog models to retrieve and query information interactively. In the near future, the challenges will mainly concern appropriate training of the domain experts. There will be a need for adequate training in technological methods that go beyond the microscope, using digital technology, the virtual environment, and AI, with all its branches and potential, and a complete understanding of the advantages and drawbacks of these technologies. The next generations of cytopathologists will certainly be more digitally adept and ready to adapt to technological change, which will facilitate their training and ability to perform tasks in the diagnostic phase.

## Figures and Tables

**Figure 1 biomedicines-11-02225-f001:**
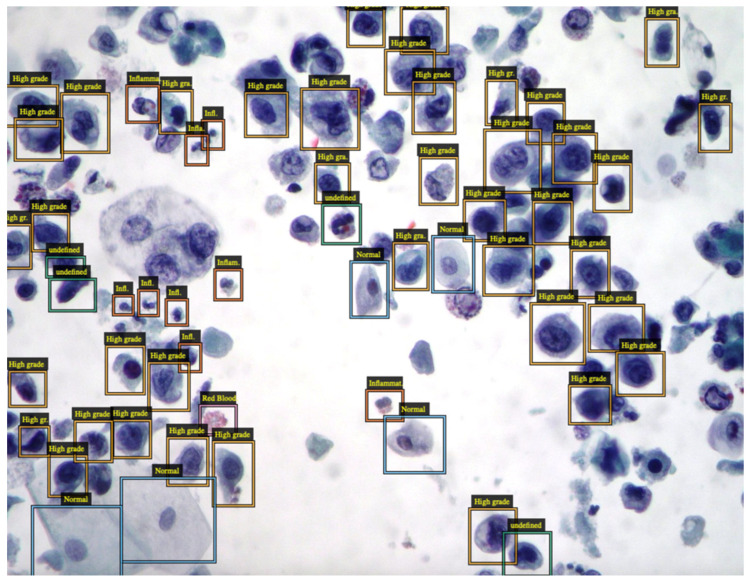
Cytopathology image of high-grade urothelial carcinoma (HGUC) showing numerous pleomorphic tumor cells. Cell detection with the addition of bounding boxes is used to annotate selected images. Each selected cell is classified according to normal or pathological features with different colors. After a suitable dataset is built and exported, a DNN can be trained to automate the process. (Conventional cytology, Papanicolaou staining, low magnification).

**Figure 2 biomedicines-11-02225-f002:**
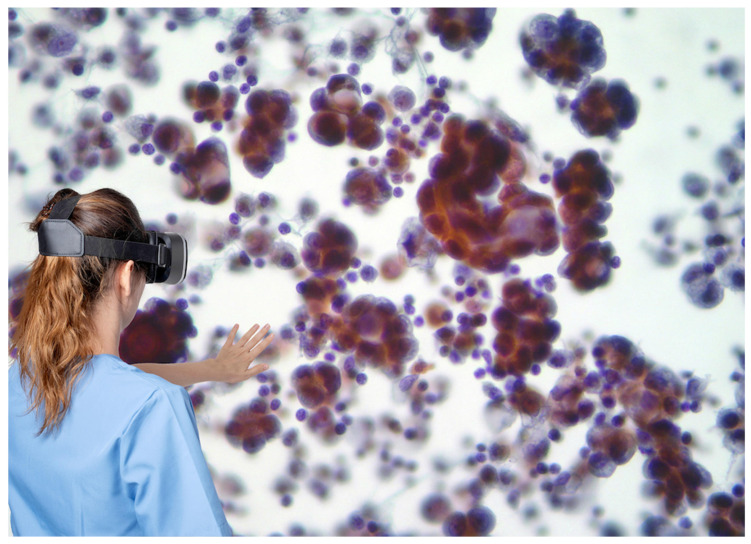
Equipped with virtual reality headsets, a clinician can visualize imaging and clinic data in an immersive fashion.

**Figure 3 biomedicines-11-02225-f003:**
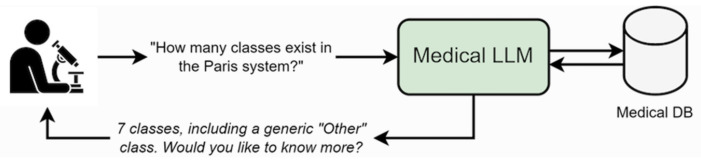
Medical chatbots and LLMs (e.g., ChatGPT) can provide interactive interfaces for querying medical data and perform literature searches.

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
