# Peer review of "Towards Artificial Intelligence Applications in Next Generation Cytopathology"

_biomedicines, 2023, doi:10.3390/biomedicines11082225_

Round 1
Reviewer 1 Report
The manuscript is a perspective on artificial intelligence as it relates to cellular pathology.
The paper is an interesting contribution and worthwhile for publication. Also, the sentences are written clearly. However, the paper requires further work on organization and structure. It is difficult to assimilate the concepts since there are very long paragraphs that should be split into multiple paragraphs that delineate the ideas. The paragraphs could also be made more concise.
Further, the concepts should be clearer in the introductory section so the reader can easily ascertain the goals of the authors.
For a few specific recommendations, the abstract section could have a couple of sentences that describe specific recommendations on use of deep learning (and advanced machine learning) in cytopathology.
Another recommendation is for the introduction section. The first paragraph could be made more concise with communication on why a cell-based approach is superior to use of tissue samples. It is probably not necessary to review all aspects of cytopathology, such as the skill level of tasks, but just the main points that are relevant to the perspective. Furthermore, the technologies could be described further in their relevance to deep learning methods.
For section 2, there is a wide lens view of the field, but it could include a clear section on segmentation. The last paragraph of this section seems the most relevant for the purpose of developing a specific perspective.
There should also be a section on cellular morphology as it relates to classification schemes such as developed by deep learning methods, even from a theoretical point of view.
Section 3 could be split into two or more subsections with clear hypotheses on deep learning approaches that depend on visual data and their relevance to the perspective. Virtual reality could be a separate subsection, too. Moreover, the network technologies could comprise a separate subsection and its advantages over existing techniques.
Section 4 could have a subsection on natural language processing in general. This is already discussed in this section, but as part of a long paragraph. The pedagogical aspects can also exist as a separate subsection, along with a critique on its impact on education and outcomes. Another subsection could be on prompt engineering since this is what is discussed later in this section. Utility as a diagnostic tool could occur as yet another subsection, along with a brief critique.
Overall, the perspective should include further details on the authors' opinions from an expert point of view and details on its impact on the discipline, including from the authors' experiences in education and the discipline. In other words, a paragraph that is a proposal is better than a sentence that generally describes the potential impact of deep learning methods. This would imprint more of the authors' expertise and opinions on the communication.
Another section could be included on how the cellular data is converted to a data format for deep learning. This can be practical or theoretical in its approach.
The manuscript is partly funded under a study of urothelial cancers. It may be useful to include a picture of a sample of this cancer at the cellular and tissue levels. These cancer cells can also be discussed in the context of currently used diagnostic tools and how they could be enhanced by deep learning, particularly in their classification capabilities that segment cancer from non-cancer cells, including specifics on where the classification schemes fail and how deep learning with a lot of data can potentially solve this problem.
The quality of word use and grammar in the English language is very good.
Author Response
Answers to reviewer #1
The paper is an interesting contribution and worthwhile for publication. Also, the sentences are written clearly.
We thank the reviewer for the positive feedback on the manuscript.
However, the paper requires further work on organization and structure. It is difficult to assimilate the concepts since there are very long paragraphs that should be split into multiple paragraphs that delineate the ideas. The paragraphs could also be made more concise.
Further, the concepts should be clearer in the introductory section so the reader can easily ascertain the goals of the authors.
We have rewritten significant portions of the manuscript (highlighted in green in the revision) to comply with these comments.
For a few specific recommendations, the abstract section could have a couple of sentences that describe specific recommendations on use of deep learning (and advanced machine learning) in cytopathology.
We have included new sentences in the abstract describing possible applications of deep learning and advanced machine learning in the field, and summarizing some specific recommendations that are expanded upon in the conclusion.
Another recommendation is for the introduction section. The first paragraph could be made more concise with communication on why a cell-based approach is superior to use of tissue samples. It is probably not necessary to review all aspects of cytopathology, such as the skill level of tasks, but just the main points that are relevant to the perspective. Furthermore, the technologies could be described further in their relevance to deep learning methods.
We have completely rewritten the first paragraphs of Section 1 to address this point. We now discuss the advantages of working at a cellular level in the first paragraph of the section, and the relevance of automating this work further on.
For section 2, there is a wide lens view of the field, but it could include a clear section on segmentation. The last paragraph of this section seems the most relevant for the purpose of developing a specific perspective.
We have rewritten the section, including a new paragraph highlighting the distinction between detection and segmentation, and how these techniques can be used in the context of digital pathology.
There should also be a section on cellular morphology as it relates to classification schemes such as developed by deep learning methods, even from a theoretical point of view.
Section 3 could be split into two or more subsections with clear hypotheses on deep learning approaches that depend on visual data and their relevance to the perspective. Virtual reality could be a separate subsection, too. Moreover, the network technologies could comprise a separate subsection and its advantages over existing techniques.
We have modified the manuscript to address this concern. Section 2 is now devoted to computer vision, highlighting the specific use cases and challenges and the problems of data acquisition and quality. Section 3 focuses exclusively on the use of AR and VR technologies in cytopathology.
Section 4 could have a subsection on natural language processing in general. This is already discussed in this section, but as part of a long paragraph. The pedagogical aspects can also exist as a separate subsection, along with a critique on its impact on education and outcomes. Another subsection could be on prompt engineering since this is what is discussed later in this section. Utility as a diagnostic tool could occur as yet another subsection, along with a brief critique.
We have divided the section into multiple sub-paragraphs following these suggestions. We have also expanded the discussion of NLP (and LLMs in particular) at the beginning of the section, and our analysis of prompt engineering in our context later.
Overall, the perspective should include further details on the authors' opinions from an expert point of view and details on its impact on the discipline, including from the authors' experiences in education and the discipline. In other words, a paragraph that is a proposal is better than a sentence that generally describes the potential impact of deep learning methods. This would imprint more of the authors' expertise and opinions on the communication.
We have expanded the manuscript to include specific points concerning the application of AI models to the field (also highlighted in green).
Another section could be included on how the cellular data is converted to a data format for deep learning. This can be practical or theoretical in its approach. The manuscript is partly funded under a study of urothelial cancers. It may be useful to include a picture of a sample of this cancer at the cellular and tissue levels. These cancer cells can also be discussed in the context of currently used diagnostic tools and how they could be enhanced by deep learning, particularly in their classification capabilities that segment cancer from non-cancer cells, including specifics on where the classification schemes fail and how deep learning with a lot of data can potentially solve this problem.
In section 2 and Figure 1 we implemented these aspects.
Reviewer 2 Report
Since this manuscript is a “perspective” rather than a conventional scientific paper I can´t comment about methods and results etc. but will give a few comments about some details in the paper.
The introduction correctly describes that PAPNET was the first approach using AI in cytopathology. And it is interesting to note that cytopathology was one of the first application fields where neural network technology was attempted. But giving that information without putting it into some context becomes misleading. Image analysis and machine learning had been applied to cytopathology since the 1950-ies starting with the huge Cytoanalyzer project in the US. And during the 70-ies and 80-ies numerous automated cytology projects were carried out in Japan, Europe and North America with similar goals and performance results as the PAPNET but based on other kinds of feature extraction and machine learning. Cytopathology, in particular screening for early detection of cervical cancer, was one of the largest early application fields of image analysis, decades before histopathology applications, probably mainly because you in cytology can deal with a small image of e.g. 256x256 pixels to analyze a cell at a time while you in histopathology need much larger images to get reasonable context, images that could not be handled until the mid 1990-ies. This lack of historical context has also influenced the Abstract.
Then it is stated that more recently AI has shown remarkable progress in… cytopathology. Without providing any references. There has been several publications including a number of review articles describing this which would be highly relevant to cite here.
In the introduction the challenging demands on cytologists are described. Then it is stated: “Institutions performing many cytological tests should be considered for training purposes.” I can´t understand how this statement fits into the context of the text. Cytologists are generally educated as part of the medical education system of universities. Should this be changed?
In the first sentences of section 2 some seemingly random papers dealing with AI in other fields than medicine are cited. I do not see the relevance.
At line 78-80 it is stated that models must be trained using large sets of images. This “would require…would mean reorganizing…” but such projects have been successfully carried out for the last 10 years so “would” is a strange choice of word. Although you are correct that the appearance of large public well characterized datasets is beginning to have and likely will have significant impact on the field.
Starting at line 124 you are describing the problems with “uncalibrated models” but it is not clear what you mean with this term. The models in literature are tested and verified in more or less ambitious ways but what exactly do you mean by “calibrated” in this context. If you refer to a specific approach a reference would be motivated.
The caption for figure 1 does not make sense. As you correctly mention at line 153 the figure shows training through supervised learning but the caption mentions “automated diagnostics and segmentation” which is contradicted by the little figure sitting at a microscope in the middle of the pipe-line.
In general I find section 3 very speculative and with limited relevance to current cytopathology, but the section deals with future trends so we will have to wait and see. You use the acronym “AR”, “VR” and “MR” without explanation even though they are increasingly used it should not be assumed that they are self-explanatory for everyone in the intended audience.
At line 193 you state that slide preparations such as thin prep and conventional cytology are generally much thicker…. But one of the motivations behind thin prep is to make the slides much thinner and easier to focus on in contrast to the thick conventional smears. Still cytology slides do pose focus problem, mainly because such high sharpness is required for registering the important information contain in the nuclear textures. Dual-view systems to handle the focus problem were introduced already in the 1970-ies not only “recently”.
I am not at all an expert on natural language processing nor on blockchain techniques so I have no comments on sections 4 and 5.
Finally there is a sentence at line 321 that I do not understand: “… witnessing a lack of vocation for diagnostic cytopathology which will have to be resolved” What does that mean?
Author Response
Answers to reviewer #2
Since this manuscript is a “perspective” rather than a conventional scientific paper I can´t comment about methods and results etc. but will give a few comments about some details in the paper.
We thank the reviewer for the thorough evaluation of the paper. We implemented most suggestions, as described better below.
The introduction correctly describes that PAPNET was the first approach using AI in cytopathology. And it is interesting to note that cytopathology was one of the first application fields where neural network technology was attempted. But giving that information without putting it into some context becomes misleading. Image analysis and machine learning had been applied to cytopathology since the 1950-ies starting with the huge Cytoanalyzer project in the US. And during the 70-ies and 80-ies numerous automated cytology projects were carried out in Japan, Europe and North America with similar goals and performance results as the PAPNET but based on other kinds of feature extraction and machine learning. Cytopathology, in particular screening for early detection of cervical cancer, was one of the largest early application fields of image analysis, decades before histopathology applications, probably mainly because you in cytology can deal with a small image of e.g. 256x256 pixels to analyze a cell at a time while you in histopathology need much larger images to get reasonable context, images that could not be handled until the mid 1990-ies. This lack of historical context has also influenced the Abstract.
We modified the abstract and included a new historical perspective in Section 1 to clarify this point, which includes a more complete review of works up to the earliest neural networks applications. We thank the reviewer for all the interesting historical pointers.
Then it is stated that more recently AI has shown remarkable progress in… cytopathology. Without providing any references. There has been several publications including a number of review articles describing this which would be highly relevant to cite here.
We have included relevant survey papers from the literature. We would be happy to include additional specific suggestions from the reviewer.
In the introduction the challenging demands on cytologists are described. Then it is stated: “Institutions performing many cytological tests should be considered for training purposes.” I can´t understand how this statement fits into the context of the text. Cytologists are generally educated as part of the medical education system of universities. Should this be changed?
We removed the sentence as it was found unclear by the reviewer.
In the first sentences of section 2 some seemingly random papers dealing with AI in other fields than medicine are cited. I do not see the relevance.
Section 2 has been completely rewritten following the suggestions of both reviewers.
At line 78-80 it is stated that models must be trained using large sets of images. This “would require…would mean reorganizing…” but such projects have been successfully carried out for the last 10 years so “would” is a strange choice of word. Although you are correct that the appearance of large public well characterized datasets is beginning to have and likely will have significant impact on the field.
We rephrased the sentence following this suggestion.
Starting at line 124 you are describing the problems with “uncalibrated models” but it is not clear what you mean with this term. The models in literature are tested and verified in more or less ambitious ways but what exactly do you mean by “calibrated” in this context. If you refer to a specific approach a reference would be motivated.
We included a more precise definition of calibration, together with specific references from the literature. Calibration in this context means that the probabilities predicted by the network do not align with their true error probability, e.g., an event predicted with a 90% probability by the network may be mistaken 25% of the time (overconfidence).
The caption for figure 1 does not make sense. As you correctly mention at line 153 the figure shows training through supervised learning but the caption mentions “automated diagnostics and segmentation” which is contradicted by the little figure sitting at a microscope in the middle of the pipe-line.
We have rewritten the caption of Figure 1 to clarify its meaning. We wanted to highlight the entire process of building a supervised learning model, from the data acquisition (left) to the training of the network (right).
In general I find section 3 very speculative and with limited relevance to current cytopathology, but the section deals with future trends so we will have to wait and see. You use the acronym “AR”, “VR” and “MR” without explanation even though they are increasingly used it should not be assumed that they are self-explanatory for everyone in the intended audience.
The acronyms have been completed.
At line 193 you state that slide preparations such as thin prep and conventional cytology are generally much thicker…. But one of the motivations behind thin prep is to make the slides much thinner and easier to focus on in contrast to the thick conventional smears. Still cytology slides do pose focus problem, mainly because such high sharpness is required for registering the important information contain in the nuclear textures. Dual-view systems to handle the focus problem were introduced already in the 1970-ies not only “recently”.
We have modified the concept.
Finally there is a sentence at line 321 that I do not understand: “… witnessing a lack of vocation for diagnostic cytopathology which will have to be resolved” What does that mean?
We have removed that sentence after rewriting parts of the section.
Round 2
Reviewer 1 Report
Dear Authors,
Thank you for the opportunity to review your latest version of the manuscript. I have additional recommendations below, too:
Line 66: should virtual reality refer to Section 3 instead of Section 4?
Line 73: please verify that the year 1957 correspond correctly to the Mellors et al reference which shows a year of 1952.
Line 77: this sentence is a bit awkward, "In the mid 1960’s, was introduced"
Line 152: should Section 4 instead be Section 5?
Line 186: should "Compare" instead be "Compared"?
Line 268: the addition could be formatted as a new paragraph instead of appended to the existing paragraph.
Line 474: should "chal- lenges" instead be "challenges"?
Line 308: "can get" can be changed to "may become".
Side note: it may be of interest to pathology to also investigate deep learning methods for image-captioning data. It is a pairing of image information with a text caption. It may help a model to learn the attributes of an image from expert and validated text descriptions, and, therefore, have an automated capacity to output textual interpretations that are putatively consistent with its training data.
Please see suggestions in the above text box.
Author Response
Line 66: should virtual reality refer to Section 3 instead of Section 4? Modified.
Line 73: please verify that the year 1957 correspond correctly to the Mellors et al reference which shows a year of 1952. Modified.
Line 77: this sentence is a bit awkward, "In the mid 1960’s, was introduced" Modified.
Line 152: should Section 4 instead be Section 5? Modified.
Line 186: should "Compare" instead be "Compared"? Modified.
Line 268: the addition could be formatted as a new paragraph instead of appended to the existing paragraph. A new paragraph has been created.
Line 474: should "chal- lenges" instead be "challenges"? Modified.
Line 308: "can get" can be changed to "may become". Modified.
Side note: it may be of interest to pathology to also investigate deep learning methods for image-captioning data. It is a pairing of image information with a text caption. It may help a model to learn the attributes of an image from expert and validated text descriptions, and, therefore, have an automated capacity to output textual interpretations that are putatively consistent with its training data. Line 268. A new reference (62) has been added regarding image-captioning including review suggestion.
